# Association of TLR-4 and TLR-9 Polymorphisms with HPV Infection and Cervical Dysplasia in Hispanic Women

**DOI:** 10.3390/cancers17233795

**Published:** 2025-11-27

**Authors:** Keimari Mendez, Ana Rosario-Santos, Magaly Martínez-Ferrer, Naydi Pérez-Rios, Alejandro O. Rivera-Torres, Josefina Romaguera, Filipa Godoy-Vitorino

**Affiliations:** 1School of Medicine, Medical Sciences Campus, University of Puerto Rico, San Juan, PR 00936, USA; ana.rosario3@upr.edu (A.R.-S.); josefina.romaguera@upr.edu (J.R.); filipa.godoy@upr.edu (F.G.-V.); 2Department of Pharmaceutical Sciences, School of Pharmacy, University of Puerto Rico, San Juan, PR 00936, USA; magaly.martinez1@upr.edu; 3University of Puerto Rico Comprehensive Cancer Center, San Juan, PR 00921, USA; alejandro.rivera17@upr.edu; 4Alliance for Clinical and Translational Research-Medical Sciences Campus, University of Puerto Rico, San Juan, PR 00921, USA; naydi.perez@upr.edu

**Keywords:** cervical dysplasia, cancer screening, human papilloma virus, cervical cancer, biomarkers

## Abstract

Cervical cancer is on the rise in Puerto Rico, even though it can often be prevented with HPV vaccines, regular Pap smears, and the early treatment of abnormal cells. This study looked at whether small genetic changes, called SNPs, in immune system genes (Toll-like receptors 4 and 9) could make women more vulnerable to HPV and cervical epithelial changes. Samples from 210 women showed that high-risk HPV was more common in those with severe abnormalities. Certain genetic variants were also associated with more advanced grades of dysplasia, reflecting cellular changes in the cervix that, if left untreated, can elevate the risk of cervical cancer. These results suggest that genetics may help explain risk and guide better cancer prevention strategies.

## 1. Introduction

Cervical cancer remains a leading cause of mortality, being the fourth most common cancer in women globally [1]. Human papilloma virus (HPV) is responsible for more than 90% of cervical cancer cases, underscoring its critical etiological role [1]. Despite ongoing advances in vaccination and screening programs, disparities in HPV-related disease outcomes remain evident across populations. Marked sociodemographic disparities and worsening patterns in both incidence and mortality among the six leading female-specific cancers emphasize the pressing need for equitable and effective prevention and intervention strategies [2]. Differences by ethnicity in HPV clearance are important topics in populational studies about cervical cancer eradication, given its importance in cancer progression [3]. In Puerto Rico (PR), HPV vaccination rates and access to screening have improved in recent years; however, Puerto Rican women remain disproportionately vulnerable compared to other ethnic groups within the United States [4,5]. Biological, environmental, and sociocultural factors likely contribute to these disparities [6]. Recent microbiome research indicates that the vaginal microbiota of Puerto Rican women tends to be more diverse and volatile [7], often characterized by a reduced abundance of protective *Lactobacillus* species [7,8,9,10]. Such microbial instability has been associated with higher susceptibility to HPV acquisition and persistence, potentially influencing the progression from infection to cervical dysplasia and cancer [11,12]. These findings highlight the importance of studying population-specific risk factors and biological mechanisms underlying HPV persistence and clearance. Among the key molecular players in the host immune response to HPV are Toll-like receptors (TLRs)—a family of pathogen recognition receptors that form a crucial component of the innate immune system [13]. TLRs are pathogen recognition receptors of the innate immune system, essential in the pathophysiology of various human diseases, including cancer and inflammation [14,15]. The signaling pathways of TLRs are engaged in the process of clearing HPV from the body [15,16]. The signal transduction pathways of TLR-4 and TLR-9 single-nucleotide polymorphisms (SNPs) have been related to HPV infection and cervical cancer [17]. Evidence suggests that TLR-9 expression increases with histopathological grade, and the HPV E6 and E7 oncoproteins deregulate the expression and function of TLR-9 [18]. A meta-analysis in 2018 found that SNP rs187084 in TLR-9 presented a significant correlation to the risk of cervical cancer [17,18,19]. Studies on genetic variations in TLRs have primarily focused on Caucasian, Chinese, and Indian populations. However, there is lack of data on Hispanic individuals, as well as on African and South American populations, despite the higher incidence of cervical cancer in these regions.

The primary goal of this study was to investigate the association between SNPs in TLR-4 and TLR-9 with cervical dysplasia severity and HPV infection. Identifying such correlations could provide novel biomarkers for individual susceptibility to severe dysplasia and cancer progression. Given the higher burden of cervical cancer among Hispanic individuals, understanding the genetic factors that contribute to HPV persistence and immune response is critical. This study highlights the importance of genetic screening in underserved populations, aiming to improve targeted prevention strategies.

## 2. Materials and Methods

### 2.1. Patient Recruitment

For this study, we used a convenience sample of 210 previously collected cervicovaginal lavage samples of a cohort of Hispanic women attending a gynecology clinic for routine cervical cancer screening or the management of abnormal results. The selected samples in this study were classified according to the Bethesda System [20], which provides a standardized framework for reporting cervical cytology findings. The categories included women with cytologic results reported as Negative for Intraepithelial Lesions or Malignancy (NILM) (n = 106), Atypical Squamous Cells of Undetermined Significance (ASC-US) (n = 28), mild dysplasia (n = 35), and severe dysplasia (n = 41). NILM represents normal cytologic findings, indicating the absence of precancerous or cancerous changes, while ASC-US refers to cells with minor abnormalities that do not clearly indicate dysplasia and usually warrant follow-up testing, such as HPV DNA analysis or repeat cytology. Mild dysplasia, corresponding to low-grade squamous intraepithelial lesions (LSILs), reflects early cellular alterations typically associated with HPV infection, which often regress spontaneously. In contrast, severe dysplasia, or high-grade squamous intraepithelial lesion (HSIL), represents more advanced precancerous changes with a higher potential for progression to cervical cancer if left untreated. The inclusion of samples across this cytologic spectrum allowed for a comprehensive evaluation of the association between genetic polymorphisms and the varying degrees of cervical epithelial abnormality. Patients were recruited as part of protocol #10510114, titled “Meta-Omic Approaches to Study Microbiome Dynamics for Cervical Cancer Prevention”, which included the SNP analyses. All patient metadata, including demographic and clinical characteristics, were de-identified and linked to samples using a unique study ID number. The classification by cervical dysplasia was negative for dysplasia, low-grade and high-grade.

### 2.2. DNA Extraction from Vaginal Lavage and HPV Genotyping

Genomic DNA was extracted from the cervicovaginal lavage samples using the Qiagen Power Soil Kit (QIAGEN LLC, Germantown Road, ML, USA) following the manufacturer’s protocol as described in Vargas-Robles et al. [7]. DNA concentration was quantified using the Qubit^®^ dsDNA HS (High Sensitivity) Assay (Waltham, MA, USA) with concentrations ranging from 5 to 100 ng/µL. For HPV detection, genomic DNA concentrations in the range of 10–30 ng/µL were used. HPV detection and genotyping were performed using the SPF10 HPV DEIA and LiPA25 version 1 (Labo Biomedical Products, Rijswijk, The Netherlands) as previously published [7,21,22]. This system employs the SFP10 PCR primer set, which amplifies a 65 bp fragment of the L1 region of the HPV genome, allowing for the detection of a broad spectrum of HPV types. The reverse primers are 5′-biotinylated, which facilitated the capture of the reverse strand on streptavidin-coated plates. The captured amplicons were then denatured with an alkaline solution, and the captured strand was identified using a digoxigenin-labeled probe mixture. The results of this DNA enzyme immunoassay (DEIA) are reported as an optical density value. The same SPF10 amplicons were used for HPV genotyping via reverse hybridization with the LiPA25 genotyping strip (version 1). Human papilloma virus (HPV) genotyping was performed using the LIPA 25 kit, which identifies both high-risk and low-risk HPV types. The high-risk HPV types detected by this assay include 16, 18, 31, 33, 34, 35, 39, 45, 51, 52, 56, 58, 59, 66, and 68/73 (shared probe). The low-risk HPV types detected include 6, 11, 40, 42, 43, 44, 53, 54, 70, and 74 and an internal control (IC) for the human β-globin gene.

### 2.3. TLR-4 and TLR-9 Genotyping

Seven SNPs were genotyped in this study: four in TLR-4 (rs4986790, rs10759931, rs11536889, rs1927911) and three in TLR-9 (rs187084, rs5743836, rs352140). Genotyping was performed using either PCR–Restriction Fragment Length Polymorphism (PCR-RFLP) or Allele-Specific PCR (AS-PCR) on previously HPV-genotyped samples: Appendix A describes the primer sequences, thermal conditions, and amplicon size for each SNP. PCRs were performed in a 25 μL volume containing 50–100 ng of genomic DNA, 0.1 mM dNTP mix, 0.1 μM of each oligonucleotide primer, and 0.8 U Taq DNA polymerase (Kapabiosystems, Wilmington, MA, USA). All the reactions were carried out on the Veriti 96 well Thermal Cycler (Applied Biosystems, Carlsbad, CA, USA). All SNPs were subjected to restriction digestion using 5 U of the respective restriction enzymes procured from New England Biolabs, USA. For the identification of SNPs by RFLP, the associated restriction enzymes, incubation temperature and time, digested products, genotypes, and mode of visualization are detailed in Appendix A. The amplified and restriction-digested products were visualized on a GelDoc system (BioRad, Hercules, CA, USA). The raw data analyzed is available as Appendix A.

### 2.4. Statistical Analysis

The frequency of alleles was determined, and we analyzed their relationship with the degree of cervical dysplasia and HPV positivity. A descriptive analysis was performed on the clinical, genotype, and sociodemographic variables. The results were reported using frequencies and percentages for the variables of interest. The sample population was stratified by the severity of dysplasia, HPV positivity, and HPV Subtype Risk. Primary analysis was performed with severity stratified into negative for intraepithelial lesion (NILM), atypical squamous cell of unknown significance (ASCUS), mild dysplasia, and severe dysplasia. Secondary analysis was performed by grouping NILM plus ASCUS into one group of no dysplasia.

For bivariate analysis, chi-square and Fisher’s exact tests were performed to independently examine the association between the variables of interest within each parameter. The significance of each variable was evaluated and reported, highlighting statistically significant associations between the variables (*p* < 0.05) based on a two-tailed test.

A multivariate analysis and logistic regression models were utilized to explore associations between cervical dysplasia and HPV while adjusting for age, smoking, severity, and HPV positivity. The results were presented with their probability value (*p*-value), strength of association (odds ratio), and respective confidence interval (95% CI).

A secondary analysis was performed, analyzing the number of alleles with a higher risk of association with cervical cancer and/or HPV infection according to the literature, with our outcomes of severe dysplasia and HPV positivity. Samples were categorized for risk as follows: no alleles, zero risk; 1 allele, low risk; 2 alleles, high risk; and 3 or more alleles, very high risk. Power estimates were performed using Gpower [23], considering the Test family (x^2^ test); Statistical Test (Goodness-of-fit test: contingency tables); and type of power analysis (Post hoc: Compute achieved power) to calculate the power (Appendix A). For the sample size, we changed the type of power analysis to a priori: compute required sample size. To calculate Cramer’s Effect Size, we used the direct estimates from the contingency tables, where we obtained Fisher’s exact test results and the *p*-value for the association between variables.

## 3. Results

### 3.1. The Demographic and Clinical Characteristics of the Study Population

We analyzed a total of 190 out of 210 cervicovaginal samples, after excluding samples with incomplete data. The cytological classifications of these samples were as follows: 106 NILM, 35 low-grade squamous intraepithelial lesion (LSIL), 41 high-grade squamous intraepithelial lesion (HSIL), and 8 ASCUS (Table 1). Of these, 61% were HPV-positive. The mean age was 38 years. The type of HPV risk was significantly associated with cervical dysplasia severity (*p* < 0.001), with high-risk HPV being the most prevalent in severe dysplasia cases, as expected (58.54%). Forty-three patients had two or more high-risk HPV genotypes present. Also, thirty patients had both high- and low-risk genotypes present (Table 1). Most of the participants were nonsmokers (91%) and non-drinkers (68%). No significant differences were found for these characteristics between the cytological classification groups (*p* > 0.05). Among the participants, 72.5% reported not having received the HPV vaccine, 13.0% reported being vaccinated, 9.8% were unsure of their vaccination status, and 4.7% had missing data (Appendix A).

### 3.2. Association of TLR4 and TLR9 Polymorphisms with Cervical Dysplasia

For TLR4 SNPs, the rs4986790 (NcoI) polymorphism showed that the AA genotype was the most prevalent, while the AG genotype was more frequent in severe dysplasia (7.32%). The rs1927911 (StyI) polymorphism revealed that the CT genotype was significantly more frequent in severe dysplasia (56.10%) (Table 1). Regarding TLR9 SNPs, the rs187084 (AfIII) polymorphism demonstrated an increase in CC genotype frequency with disease severity, rising from 13.76% in NILM to 29.27% in severe dysplasia. The CT heterozygous genotype was more common in severe dysplasia (34.15%). Similarly, the rs5743836 (BstNI) polymorphism indicated that the TC genotype was the most frequent in severe dysplasia (48.78). High levels of TLR4 expression were significantly associated with severe dysplasia (62.50%). For TLR9 expression, high levels were found in 35% of severe dysplasia cases, while very high levels were observed in 17.5% of cases (Table 1). Other polymorphisms were also evaluated [e.g., TLR4 SNP: rs10759931 (KpnI), rs11536889 (Earl); TLR9 SNP: rs352140 (BstUI)], but they did not show a significant difference with the cervical phenotype groups (*p* > 0.05).

### 3.3. Association of TLR4 and TLR9 Polymorphisms with TLR Expression Levels

We further stratified our results into groups representing very low, low, high, and very high levels of TLR genotypes. Table 2 presents the association between TLR-4 expression levels (categorized as low, high, and very high) and various clinical and demographic factors. Sociodemographic characteristics, such as age, BMI, pregnancy, drinking status, and smoking status, were not found to be different among TLR-4 expression groups (*p* > 0.05). The analysis of specific SNPs in TLR-4 revealed several significant connections. The rs10759931 (KpnI) polymorphism showed a highly significant association with TLR-4 expression (*p* < 0.001). Individuals with the GG genotype were the most likely to exhibit very high TLR-4 expression levels (71.43%), while the AA genotype was primarily observed in those with high expression levels (26.42%). The rs4986790 (NcoI) polymorphism also demonstrated a strong relationship (*p* < 0.001), where the AA genotype was almost exclusively found in individuals with low TLR-4 expression levels (100%), while the AG genotype was prevalent among those with very high TLR-4 expression levels (92.86%). The rs1927911 (StyI) polymorphism showed another highly significant association (*p* < 0.001), with the CT genotype being common in individuals with very high TLR-4 expression levels (85.71%) (Table 2).

We examine the association between TLR-9 expression levels, categorized as very low, low, high, and very high, and various clinical and demographic factors (Table 3). Sociodemographic characteristics, such as age, BMI, pregnancy, drinking status, and smoking status, were not found to be different among TLR-9 expression groups (*p* > 0.05). Similarly, no statistically significant difference was observed between TLR-9 expression HPV status (*p*-value = 0.130) and the type of HPV-related risk (categorized as none, low, high, or both low and high) (*p*-value = 0.396). However, analyzing specific SNPs in TLR-9 revealed highly significant associations (Table 3). The rs187084 (AflII) polymorphism was strongly related to TLR-9 expression, with a *p*-value of less than 0.001. The CC genotype was more prevalent in those with low and high TLR-9 expression levels but was absent in individuals with very high TLR-9 expression levels. The CT genotype was particularly common in individuals with high (50.70%) and very high (100%) expression levels. The rs5743836 (BstNI) polymorphism also demonstrated a highly significant association (*p* < 0.001), where the TT genotype was predominant in individuals with very low TLR-9 expression levels (92.31%) but absent in individuals with very high expression levels. In contrast, the TC genotype was the most frequently observed in those with high and very high TLR-9 expression levels. Another SNP, rs352140 (BstUI), also had a *p*-value of less than 0.001. The GG genotype was absent in individuals with very low TLR-9 expression levels but was present in those with low and high expression levels. The GA genotype was strongly associated with high and very high expression levels (Table 3). The heterozygous polymorphisms in affii and BstNI are associated with very high levels of TLR-9.

### 3.4. Risk of Cervical Dysplasia Associated with TLR4 Polymorphisms and HPV Infections

We found that the rs4986790 (NcoI) polymorphism in TLR-4 was significantly associated with an increased risk of cervical dysplasia (Table 4). Women with the AA genotype had an approximately threefold higher odds of having cervical dysplasia compared to those with the AG genotype. The unadjusted odds ratio was 3.11 (95% CI: 1.10–8.78, *p* ≤ 0.05), while the adjusted models showed similar results with odds ratios of 3.06 (1.08–8.72) and 2.66 (0.91–7.77). Although slightly reduced in the fully adjusted model, the association remains strong. HPV infection was also analyzed as a risk factor for cervical dysplasia. High-risk HPV infection showed an increased risk in the unadjusted model (OR: 1.75; 95% CI: 0.98–3.14), but after adjusting for confounders, the odds ratio decreased and was no longer statistically significant (OR: 0.68; 95% CI: 0.20–2.31) (Table 4).

We also examined the odds ratios and confidence intervals for HPV positivity in relation to the rs11536889 (EarI) polymorphism in TLR-4, assessing both unadjusted and adjusted models. For the CC genotype, the unadjusted odds ratio was 3.17 (95% CI: 0.34–29.12), which remained elevated but was not statistically significant after adjustments for age, smoking, and severity, with an odds ratio of 3.48 (95% CI: 0.38–32.29) (Table 5). The CG genotype, however, showed a statistically significant association with HPV positivity. The unadjusted odds ratio was 2.15 (95% CI: 1.06–4.35, *p* ≤ 0.05), indicating that individuals with this genotype were more than twice as likely to be HPV-positive compared to those with the GG genotype. This association remained significant in both adjusted models, with odds ratios of 2.18 (95% CI: 1.07–4.43) and 2.15 (95% CI: 1.05–4.38) (Table 5), reinforcing the role of the CG genotype in HPV susceptibility. Overall, the findings suggest that the rs11536889 (EarI) polymorphism in TLR-4 is associated with an increased risk of HPV positivity, particularly for individuals carrying the CG genotype (Table 5).

## 4. Discussion

Our main objective was to explore the correlation of four SNPs of TLR 4 and three SNPs of TLR-9 with the presence of cervical dysplasia and HPV infection in the cervicovaginal lavages of women attending a gynecology clinic. This study provides the first evidence of an association between specific SNPs in TLR4 and TLR9 and the severity of cervical dysplasia and HR-HPV infection in a cohort of Hispanic women from Puerto Rico. The most important finding was that four polymorphisms had significant associations with the risk of cervical dysplasia and HPV high-risk infection in our sample. For TLR-4, the rs4986790 (NcoI) polymorphism, which causes an amino acid exchange from aspartate to glycine (AA/AG), showed that individuals with the homozygous AA genotype had a significantly higher risk of cervical dysplasia than those with the AG genotype, with odds ratios suggesting a threefold increase in risk. In a study of Tunisian women with cervical cancer compared with controls, the TLR-4 polymorphism Asp299Gly (rs4986790) was found to be associated with a higher risk of cervical cancer as well [19]. In a second Tunisian case–control study with 130 cervical cancer patients and 260 controls, it was shown that the rs4986790 dominant genotype Asp/Asp was significantly more frequent among cervical cancer cases in the early stages (I + II) and advanced stages (III + IV) than controls [24]. Pandey et al., in a study of North Indian women, did observe an association with cervical cancer risk at the genotype, allele, and haplotype levels [25]. Differences among ethnicities suggests that there is a need for specific tests for certain high-risk populations. For TLR-9, the rs187084 (AfIII) and rs5743836 (BstNI) polymorphisms showed that individuals with heterozygous CT and TC genotypes, respectively, were more likely to have high or very high levels of TLR-9 expression, which has been linked to increased immune response activation [26]. In contrast, homozygous CC genotypes in rs187084 were associated with lower TLR-9 expression levels, potentially weakening immune defenses against HPV [27]. Similarly, homozygous TT genotypes in rs5743836 were predominantly observed in individuals with lower TLR-9 expression levels, suggesting a role in HPV persistence [28,29]. Our data supports the hypothesis that genetic variations in TLR-4 may influence immune responses to HPV, potentially affecting viral persistence and infection outcomes. The CC genotype also showed an increased risk, though the wide confidence interval suggests that more data is needed to confirm its role. These results highlight the potential significance of TLR-4 polymorphisms as biomarkers for HPV susceptibility and cervical dysplasia risk. The mixture of races that characterizes our population can impact our analysis. A total of 3% of women were from the Dominican Republic in our sample (compared to 2% according to data of statistical institute of PR from 2015 to 2019). Puerto Rican and Dominican individuals are genetically similar, both sharing a large proportion of African ancestry (20–40%) and similar proportion of European (50–60) and native ancestry (5–15%) [30].

Our study offers several notable strengths. These include a well-characterized clinical cohort, a comprehensive analysis of multiple SNPs in two key TLR genes, and a focus on a high-risk and understudied population. The inclusion of a relatively large sample of 210 patients, all recruited from the same gynecology clinic and handled by the same research personnel, ensured consistency throughout this study. All laboratory analyses were performed in a single facility, minimizing methodological variability, such as interlaboratory differences. Additionally, all specimens were collected by experienced gynecologists, which optimized sample quality and facilitated efficient DNA extraction from cervicovaginal sites. Another strength is the ongoing nature of this study, which will continue to collect and store additional samples in a biorepository for future research.

However, several limitations should be acknowledged. Although the sample size was adequate for the primary analyses, it may have limited our ability to detect more subtle genetic associations. Moreover, the cross-sectional design precludes the establishment of causal relationships. The absence of cancer cases and incomplete typification of certain HPV types also restricted the scope of our analysis. In particular, some HPV-positive samples contained serotypes not detected by the testing kits used, preventing a comprehensive evaluation of all HPV serotypes present in our cohort.

Future longitudinal studies with larger sample sizes are warranted to validate these findings and to further elucidate the roles of these and other genetic factors in the natural history of HPV infection and the development of cervical dysplasia. The investigation of genetic variants such as the *TLR4* and *TLR9* polymorphisms is justified because these innate immune receptors play a critical role in the body’s first line of defense against viral infections, including HPV. Variations in these genes may alter immune signaling pathways, influencing the persistence of HPV infection and the progression from infection to dysplasia and cancer. Understanding these associations provides valuable insights into the mechanisms underlying cervical carcinogenesis. From a clinical and public health standpoint, studying these variants can contribute to risk stratification; however, in our case, we acknowledge the limitation of having a smaller sample size. With the effect size obtained from the results of this study, we would need approximately 400 patients to achieve a power of 80% using an alpha level of 0.05. Identifying individuals with genetic profiles that confer increased susceptibility could allow for more personalized preventive strategies, such as tailored screening intervals or combined testing approaches (Pap smear and HPV co-testing) in those at elevated risk. Although current evidence is not sufficient to modify established screening guidelines, these findings represent an important first step toward integrating molecular and genetic data into preventive healthcare.

## 5. Conclusions

Our study provides novel evidence that specific SNPs in TLR4 and TLR9 are associated with an increased risk of cervical dysplasia and HR-HPV infection in our patients. The homozygous AA genotype in rs4986790 was linked to a higher likelihood of cervical dysplasia, while the heterozygous CG genotype in rs11536889 was associated with greater susceptibility to HPV infection. These results suggest that variations in TLR-4 may impair the immune recognition of HPV, facilitating persistent infection and cellular changes leading to dysplasia. Individuals with heterozygous CT and TC genotypes for TLR-9 (rs187084 (AfIII) and rs5743836 (BstNI)) were more likely to exhibit high TLR-9 expression levels, indicating a heightened immune response. Conversely, homozygous CC and TT genotypes were associated with lower TLR-9 expression levels, suggesting reduced immune activation and a potential failure to clear HPV infection, which could increase the risk of persistent infection and disease progression. These findings suggest that TLR polymorphisms could serve as potential biomarkers for identifying individuals at higher risk for cervical dysplasia and HPV-related malignancies, allowing for risk stratification in susceptible populations. Individualized treatment according to risk data provided by these biomarkers could improve the management of Hispanic women and other susceptible populations for HPV-related malignancies. A high-risk profile can direct clinical management to a more aggressive approach when the biomarkers are present. Future studies with larger samples of patients with cervical dysplasia and cancer are needed to confirm our findings in advanced stages of the disease.

## Figures and Tables

**Table 1 cancers-17-03795-t001:** Association between demographic and clinical characteristics and cervical phenotype groups.

Selected Characteristics	Cervical Phenotype				*p*-Value ^1^
	NILM n = 106	ASCUS n = 8	Mild Dysplasia n = 35	Severe Dysplasia n = 41	
**HPV Status**, n (%)					
Negative	50 (45.87)	3 (37.50)	8 (22.86)	14 (34.15)	0.138
Positive	56 (51.38)	5 (62.50)	27 (77.14)	27 (65.85)	
Unknown	3 (2.75)	0 (0.00)	0 (0.00)	0 (0.00)	
**Type of Risk** ^a^, n (%)					<0.001
None	50 (45.87)	3 (37.50)	8 (22.86)	14 (34.15)	
Low-risk HPV	5 (4.59)	5 (62.50)	2 (5.71)	0 (0.00)	
High-risk HPV	32 (29.36)	0 (0.00)	17 (48.57)	24 (58.54)	
Both low and high	19 (17.43)	0 (0.00)	8 (22.86)	3 (7.32)	
**TLR4: rs4986790 (NcoI)** ^b^, n (%)					0.221
AA	89 (81.65)	7 (87.50)	33 (94.29)	37 (90.24)	
AG	18 (16.51)	0 (0.00)	2 (5.71)	3 (7.32)	
**TLR4: rs1927911 (StyI)** ^b^, n (%)					0.107
CC	13 (11.93)	2 (25.00)	8 (22.86)	3 (7.32)	
CT	42 (38.53)	1 (12.50)	13 (37.14)	23 (56.10)	
TT	52 (47.71)	4 (50.00)	14 (40.00)	14 (34.15)	
**TLR9: rs187084 (AflII)** ^b^, n (%)					0.134
CC	15 (13.76)	0 (0.00)	6 (17.14)	12 (29.27)	
CT	52 (47.71)	2 (25.00)	12 (34.29)	14 (34.15)	
TT	40 (36.70)	5 (62.50)	17 (48.57)	14 (34.15)	
**TLR9: rs5743836 (BstNI)** ^b^, n (%)					0.321
TT	43 (39.45)	1 (12.50)	11 (31.43)	14 (34.15)	
TC	56 (51.38)	6 (75.00)	23 (65.71)	20 (48.78)	
CC	8 (7.34)	0 (0.00)	1 (2.86)	6 (14.63)	
**TLR-4**, n (%)					0.418
Low	35 (32.71)	5 (71.43)	16 (45.71)	13 (32.50)	
High	62 (57.94)	2 (28.57)	17 (48.57)	25 (62.50)	
Very high	10 (9.35)	0 (0.00)	2 (5.71)	2 (5.00)	
**TLR-9**, n (%)					0.533
Very low	17 (15.89)	0 (0.00)	4 (11.43)	5 (12.50)	
Low	27 (25.23)	4 (57.14)	7 (20.00)	14 (35.00)	
High	37 (34.58)	3 (42.86)	17 (48.57)	14 (35.00)	
Very high	26 (24.30)	0 (0.00)	7 (20.00)	7 (17.50)	

^1^ *p*-values were calculated using Fisher’s exact test. Missing values: ^a^ = 3; ^b^ = 4.

**Table 2 cancers-17-03795-t002:** Association between selected characteristics and TLR-4 levels.

Selected Characteristics	TLR-4			*p*-Value ^1^
	Low n = 69	High n = 106	Very High n = 14	
**HPV Status**, n (%)				0.543
Negative	31 (44.93)	35 (33.02)	6 (42.86)	
Positive	37 (53.62)	69 (65.09)	8 (57.14)	
Unknown	1 (1.45)	2 (1.89)	0 (0.00)	
**Type of Risk**, n (%)				0.573
None	31 (44.93)	35 (33.02)	6 (42.86)	
Low	6 (8.70)	6 (5.66)	0 (0.00)	
High	20 (28.99)	46 (43.40)	7 (50.00)	
Both Low and High	11 (15.94)	17 (16.04)	1 (7.14)	
Missing	1 (1.45)	2 (1.89)	0 (0.00)	
**TLR4: rs10759931 (KpnI)** ^a^, n (%)				<0.001
AA	0 (0.00)	28 (26.42)	3 (21.43)	
AG	38 (55.07)	46 (43.40)	1 (7.14)	
GG	31 (44.93)	32 (30.19)	10 (71.43)	
**TLR4:rs11536889 (EarI)** ^a^, n (%)				0.109
CC	1 (1.45)	4 (3.77)	0 (0.00)	
CG	14 (20.29)	37 (34.91)	2 (14.29)	
GG	54 (78.26)	65 (61.32)	12 (85.71)	
**TLR4: rs4986790 (NcoI)** ^a^, n (%)				<0.001
AA	69 (100.00)	96 (90.57)	1 (7.14)	
AG	0 (0.00)	10 (9.43)	13 (92.86)	
**TLR4: rs1927911 (StyI)** ^a^, n (%)				<0.001
CC	24 (34.78)	2 (1.89)	0 (0.00)	
CT	0 (0.00)	67 (63.21)	12 (85.71)	
TT	45 (65.22)	37 (34.91)	2 (14.29)	

^1^ *p*-values were calculated using Fisher’s exact test.  Missing values: ^a^ = 4.

**Table 3 cancers-17-03795-t003:** Association between selected characteristics and TLR-9.

Selected Characteristics	TLR-9				*p*-Value ^1^
	Very Low n = 26	Low n = 52	High n = 71	Very High n = 40	
**HPV Status**, n (%)					0.130
Negative	13 (50.00)	24 (46.15)	23 (32.39)	12 (30.00)	
Positive	12 (46.15)	27 (51.92)	47 (66.20)	28 (70.00)	
Unknown	1 (3.85)	1 (1.92)	1 (1.41)	0 (0.00)	
**Type of Risk**, n (%)					0.396
None	13 (50.00)	24 (46.15)	23 (32.39)	12 (30.00)	
Low	0 (0.00)	3 (5.77)	7 (9.86)	2 (5.00)	
High	9 (34.62)	15 (28.85)	31 (43.66)	18 (45.00)	
Both Low and High	3 (11.54)	9 (17.31)	9 (12.68)	8 (20.00)	
Missing	1 (3.85)	1 (1.92)	1 (1.41)	0 (0.00)	
**TLR9: rs187084 (AflII)** ^a^, n (%)					<0.001
CC	3 (11.54)	14 (26.92)	16 (22.54)	0 (0.00)	
CT	0 (0.00)	4 (7.69)	36 (50.70)	40 (100.00)	
TT	23 (88.46)	34 (65.38)	19 (26.76)	0 (0.00)	
**TLR9: rs5743836 (BstNI)** ^a^, n (%)					<0.001
TT	24 (92.31)	21 (40.38)	24 (33.80)	0 (0.00)	
TC	0 (0.00)	23 (44.23)	42 (59.15)	40 (100.00)	
CC	2 (7.69)	8 (15.38)	5 (7.04)	0 (0.00)	
**TLR9: rs352140 (BstUI)**, n (%)					<0.001
GG	0 (0.00)	15 (28.85)	14 (19.72)	11 (27.50)	
GA	0 (0.00)	11 (21.15)	50 (70.42)	29 (72.50)	
AA	26 (100.00)	26 (50.00)	7 (9.86)	0 (0.00)	

^1^ *p*-values were calculated using Fisher’s exact test. Missing values: ^a^ = 4.

**Table 4 cancers-17-03795-t004:** Odds ratios (95% confidence intervals) ^1^ for cervical dysplasia, according to risk factors (n = 193).

Dysplasia	Unadjusted	Adjusted	
		Model 1 ^2^	Model 2 ^3^
	OR (95% CI)	OR (95% CI)	OR (95% CI)
**TLR4: rs4986790 (NcoI)**			
AA	3.11 (1.10–8.78) *	3.06 (1.08–8.72) *	2.66 (0.91–7.77)
AG	Reference	Reference	Reference
**HPV Risk**			
High	1.75 (0.98–3.14)	1.74 (0.96–3.13)	0.68 (0.20–2.31)
None/Low	Reference	Reference	Reference

^1^ Estimates were obtained from logistic regression models for the binary cervical dysplasia outcome (yes/no). ^2^ Model 1 was adjusted for age and smoking (smoker/nonsmoker). ^3^ Model 2 was adjusted for age, smoking (smoker/nonsmoker), and HPV positivity (yes/no). * Statistically significant results: *p*-value ≤ 0.05.

**Table 5 cancers-17-03795-t005:** Odds ratios (95% confidence intervals) ^1^ for HPV positivity, according to EARI expression (n = 115).

Positive HPV	Unadjusted	Adjusted	
		Model 1 ^2^	Model 2 ^3^
	OR (95% CI)	OR (95% CI)	OR (95% CI)
**TLR4:rs11536889 (EarI)**			
CC	3.17 (0.34–29.12)	3.28 (0.36–30.29)	3.48 (0.38–32.29)
CG	2.15 (1.06–4.35) *	2.18 (1.07–4.43) *	2.15 (1.05–4.38) *
GG	Reference	Reference	Reference

^1^ Estimates were obtained from logistic regression models for the binary cervical dysplasia outcome (yes/no). ^2^ Model 1 was adjusted for age and smoking (smoker/nonsmoker). ^3^ Model 2 was adjusted for age, smoking (smoker/nonsmoker), and severity. * Statistically significant results: *p*-value ≤ 0.05.

## Data Availability

All data is provided in this manuscript as Appendix A.

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
