# Peer review of "Association of TLR-4 and TLR-9 Polymorphisms with HPV Infection and Cervical Dysplasia in Hispanic Women"

_cancers, 2025, doi:10.3390/cancers17233795_

Round 1

Reviewer 1 Report

Comments and Suggestions for Authors

This is an interesting study. However, it is not clear the usefulness of investigate these genetic variants in terms of changing strategies. What you propose, a more frequent screening in these populations, always combining Pap smear with HPV check, etc? Some related comments are necessary. 

Author Response

Comment Reviewer 1:

This is an interesting study. However, it is not clear the usefulness of investigate these genetic variants in terms of changing strategies. What you propose, a more frequent screening in these populations, always combining Pap smear with HPV check, etc? Some related comments are necessary. 

Response: The investigation of genetic variants such as TLR4 and TLR9 polymorphisms is justified because these innate immune receptors play a critical role in the body’s first line of defense against viral infections, including HPV. Variations in these genes may alter immune signaling pathways, influencing the persistence of HPV infection and the progression from infection to dysplasia and cancer. Understanding these associations provides valuable insights into the mechanisms underlying cervical carcinogenesis. From a clinical and public health standpoint, studying these variants can contribute to risk stratification; however, in our case, we acknowledge the limitation of having a lower sample size. Identifying individuals with genetic profiles that confer increased susceptibility could allow for more personalized preventive strategies, such as tailored screening intervals or combined testing approaches (Pap smear and HPV co-testing) in those at elevated risk. Although current evidence is not sufficient to modify established screening guidelines, these findings represent an important first step toward integrating molecular and genetic data into preventive healthcare. We have now added this explanation to the discussion.

Reviewer 2 Report

Comments and Suggestions for Authors

Review of article “Association of TLR-4 and TLR-9 Polymorphisms with HPV Infection and
Cervical Dysplasia in Hispanic Women”.

Cervical cancer and Human Papillomavirus (HPV) infection represent major public health concerns worldwide, with significant social and economic implications. Despite advances in vaccination and screening, disparities in HPV infection rates and cervical cancer incidence persist across populations. Understanding the biological, genetic, and environmental factors influencing HPV persistence and cancer progression is essential for developing targeted prevention strategies and reducing health inequities.

The authors proposed an appropriate and well-founded research strategy aimed at understanding the genetic factors influencing HPV persistence and cervical dysplasia severity. By focusing on SNPs in TLR-4 and TLR-9 and their potential role as biomarkers, the study offers valuable insights that could enhance targeted prevention and improve health outcomes in underserved Hispanic populations.

In summary, the study demonstrates that specific polymorphisms — TLR4 rs4986790 (AA genotype), TLR4 rs11536889 (CG genotype), TLR9 rs187084 (CT genotype), and TLR9 rs5743836 (TC genotype) — are significantly associated with increased risk of cervical dysplasia and HPV infection, suggesting their potential value as genetic biomarkers for risk assessment and personalized management.

Could the authors please specify which HPV -high risk genotypes were identified among the patients included in the study? Providing this information would enhance the interpretation of the results and clarify whether certain HPV types are more strongly associated with the observed TLR polymorphisms in this geographical region.

Could the authors clarify whether the patients included in the study had received HPV vaccination prior to their enrolment?

I would suggest revising the graphical presentation of the results. Instead of using long tables, it might be more effective to highlight only the most relevant findings or, if possible, to present the data using figures or charts to improve clarity and visual impact.

Author Response

Comment Reviewer 2:

-Could the authors please specify which HPV -high risk genotypes were identified among the patients included in the study? Providing this information would enhance the interpretation of the results and clarify whether certain HPV types are more strongly associated with the observed TLR polymorphisms in this geographical region.

Response: We have now added a supplementary table with the description of the HPV risk types (metadata)

-Could the authors clarify whether the patients included in the study had received HPV vaccination prior to their enrolment? Response: We had some vaccinated and unvaccinated participants and other that didn’t knew the information. We will be able to provide the exact percentages but was not possible to be obtained for today.

-I would suggest revising the graphical presentation of the results. Instead of using long tables, it might be more effective to highlight only the most relevant findings or, if possible, to present the data using figures or charts to improve clarity and visual impact.

Response: We appreciate the comment and have now reduced the table sizes.

Reviewer 3 Report

Comments and Suggestions for Authors

This novel study on the correlation of SNPs in TLR4 and TLR9  to HPV infection is well designed and the statistical tests employed are acceptable. The whole analysis is based on cervico-vaginal samples drawn from a Hispanic patients. Of the 210 samples used in the study 190 were employed in the final analysis. The study design is clear and well written. The conclusions drawn from the analysis indicates a gross correlation of 61% . 
The authors could improve the manuscript by addressing the following:

  1. Indicate how the sample calculation was done for useful statistical analysis.
  2. Clarify -this is a part of a protocol # 105101144 -Meta-OMIC Approaches to study microbiome dynamics for cancer cervix prevention. Was this study approved by the IRB?
  3. Suggest the significance of the results in moving forwards in use of SNPs in TLR4 , TLR9 and other markers in this area.
  4. Include specifics about the classification of LGSIL HGSIL and genotypes of HPV studied.

Author Response

Comment Reviewer 3:

The authors could improve the manuscript by addressing the following:

1. Indicate how the sample calculation was done for useful statistical analysis.

Response: The use of a convenience sample may have limited generalizability of the results and reduce the power to detect subtle association. Nonetheless, the sample was deemed adequate for the primary analyses allowing the collection of essential data that has not been previously explored in this population

2. Clarify -this is a part of a protocol # 105101144 -Meta-OMIC Approaches to study microbiome dynamics for cancer cervix prevention. Was this study approved by the IRB?

Response: Our study received additional IRB approval for the SNPs analysis of the previously collected samples-Protocol Number: B3550122. It was clarified in the text.

3. Suggest the significance of the results in moving forwards in use of SNPs in TLR4 , TLR9 and other markers in this area.

Response: We have now added a statement at the end of the discussion highlighting the usefulness of these 2 markers.

4. Include specifics about the classification of LGSIL HGSIL and genotypes of HPV studied.

Response: We have added this explanation to the patient recruitment section in materials and methods.